# SLAPS: Self-Supervision Improves Structure Learning for Graph Neural Networks

**Bahare Fatemi**[*]
University of British Columbia
bfatemi@cs.ubc.ca

**Layla El Asri**
Borealis AI
layla.elasri@borealisai.com

**Seyed Mehran Kazemi**[*]
Google Research
mehrankazemi@google.com

## Abstract

Graph neural networks (GNNs) work well when the graph structure is provided. However, this structure may not always be available in real-world applications. One solution to this problem is to infer a task-specific latent structure and then apply a GNN to the inferred graph. Unfortunately, the space of possible graph structures grows super-exponentially with the number of nodes and so the task-specific supervision may be insufficient for learning both the structure and the GNN parameters. In this work, we propose the **S**imultaneous **L**earning of **A**djacency and GNN **P**arameters with **S**elf-supervision, or SLAPS, a method that provides more supervision for inferring a graph structure through self-supervision. A comprehensive experimental study demonstrates that SLAPS scales to large graphs with hundreds of thousands of nodes and outperforms several models that have been proposed to learn a task-specific graph structure on established benchmarks.

## 1 Introduction

Graph representation learning has grown rapidly and found applications in domains where a natural graph of the data points is available [4, 25]. Graph neural networks (GNNs) [40] have been a key component to the success of the research in this area. Specifically, GNNs have shown promising results for semi-supervised classification when the available graph structure exhibits a high degree of homophily (i.e. connected nodes often belong to the same class) [57].

We study the applicability of GNNs to (semi-supervised) classification problems where a graph structure is *not* readily available. The existing approaches for this problem either fix a similarity graph between the nodes or learn the GNN parameters and a graph structure simultaneously (see Related Work). In both cases, one main goal is to construct or learn a graph structure with a high degree of homophily with respect to the labels to aid the GNN classification. The latter approach is sometimes called *latent graph learning* and often results in higher predictive performance compared to the former approach (see, e.g., [12]).

We identify a supervision starvation problem in latent graph learning approaches in which the edges between pairs of nodes that are far from labeled nodes receive insufficient supervision; this results in learning poor structures away from labeled nodes and hence poor generalization. We propose a solution for this problem by adopting a multi-task learning framework in which we supplement the classification task with a self-supervised task. The self-supervised task is based on the hypothesis

---

[*]Work was done when authors were at Borealis AI.

35th Conference on Neural Information Processing Systems (NeurIPS 2021).

that a graph structure that is suitable for predicting the node features is also suitable for predicting the node labels. It works by masking some input features (or adding noise to them) and training a separate GNN aiming at updating the adjacency matrix in such a way that it can recover the masked (or noisy) features. The task is generic and can be combined with several existing latent graph learning approaches.

We develop a latent graph learning model, dubbed SLAPS, that adopts the proposed self-supervised task. We provide a comprehensive experimental study on nine datasets (thirteen variations) of various sizes and from various domains and perform thorough analyses to show the merit of SLAPS.

Our main contributions include: 1) identifying a supervision starvation problem for latent graph learning, 2) proposing a solution for the identified problem through self-supervision, 3) developing SLAPS, a latent graph learning model that adopts the self-supervised solution, 4) providing comprehensive experimental results showing SLAPS substantially outperforms existing latent graph learning baselines from various categories on various benchmarks, and 5) providing an implementation for latent graph learning that scales to graphs with hundreds of thousands of nodes.

## 2 Related work

Existing methods that relate to this work can be grouped into the following categories. We discuss selected work from each category and refer the reader to [60] for a full survey.

**Similarity graph:** One approach for inferring a graph structure is to select a similarity metric and set the edge weight between two nodes to be their similarity [39, 44, 3]. To obtain a sparse structure, one may create a kNN similarity graph, only connect pairs of nodes whose similarity surpasses some predefined threshold, or do sampling. As an example, in [14] a (fixed) kNN graph using the cosine similarity of the node features is created. In [47], this idea is extended by creating a fresh graph in each layer of the GNN based on the node embedding similarities in that layer. Instead of choosing a single similarity metric, in [15] several (potentially weak) measures of similarity are fused. The quality of the predictions of these methods depends heavily on the choice of the similarity metric(s).

**Fully connected graph:** Another approach is to start with a fully connected graph and assign edge weights using the available meta-data or employ the GNN variants that provide weights for each edge via an attention mechanism [45, 53]. This approach has been used in computer vision [e.g., 43], natural language processing [e.g., 56], and few-shot learning [e.g., 13]. The complexity of this approach grows rapidly making it applicable only to small-sized graphs. Zhang et al. [54] propose to define local neighborhoods for each node and only assume that these local neighborhoods are fully connected. Their approach relies on an initial graph structure to define the local neighborhoods.

**Latent graph learning:** Instead of a similarity graph based on the initial features, one may use a graph generator with learnable parameters. In [30], a fully connected graph is created based on a bilinear similarity function with learnable parameters. In [12], a Bernoulli distribution is learned for each possible edge and graph structures are created through sampling from these distributions. In [49], the input structure is updated to increase homophily based on the labels and model predictions. In [6], an iterative approach is proposed that iterates over projecting the nodes to a latent space and constructing an adjacency matrix from the latent representations multiple times. A common approach in this category is to learn a projection of the nodes to a latent space where node similarities correspond to edge weights or edge probabilities. In [48], the nodes are projected to a latent space by learning weights for each of the input features. In [38, 21, 8], a multi-layer perceptron is used for projection. In [52, 55], a GNN is used for projection; it uses the node features and an initial graph structure. In [26], different graph structures are created in different layers by using separate GNN projectors, where the input to the GNN projector in a layer is the projected values and the generated graph structure from the previous layer. In our experiments, we compare with several approaches from this category.

**Leveraging domain knowledge:** In some applications, one may leverage domain knowledge to guide the model toward learning specific structures. For example, in [24], abstract syntax trees and regular languages are leveraged in learning graph structures of Python programs that aid reasoning for downstream tasks. In [23], the structure learning is guided for robustness to adversarial attacks through the domain knowledge that clean adjacency matrices are often sparse and low-rank and exhibit feature smoothness along the connected nodes. Other examples in this category include

[19, 38]. In our paper, we experiment with general-purpose datasets without access to domain knowledge.

**Proposed method:** Our model falls within the latent graph learning category. We supplement the training with a self-supervised objective to increase the amount of supervision in learning a structure. Our self-supervised task is inspired by, and similar to, the pre-training strategies for GNNs [17, 18, 22, 51, 58] (specifically, we adopt the multi-task learning framework of You et al. [51]), but it differs from this line of work as we use self-supervision for learning a graph structure whereas the above methods use it to learn better (and, in some cases, transferable) GNN parameters.

## 3   Background and notation

We use lowercase letters to denote scalars, bold lowercase letters to denote vectors and bold uppercase letters to denote matrices. $I$ represents an identity matrix. For a vector $v$, we represent its $i^{\text{th}}$ element as $v_i$. For a matrix $M$, we represent the $i^{\text{th}}$ row as $M_i$ and the element at the $i^{\text{th}}$ row and $j^{\text{th}}$ column as $M_{ij}$. For an attributed graph, we use $n$, $m$ and $f$ to represent the number of nodes, edges, and features respectively, and denote the graph as $\mathcal{G} = \{\mathcal{V}, A, X\}$ where $\mathcal{V} = \{v_1, \dots, v_n\}$ is a set of nodes, $A \in \mathbb{R}^{n \times n}$ is an adjacency matrix with $A_{ij}$ indicating the weight of the edge from $v_i$ to $v_j$ ($A_{ij} = 0$ implies no edge), and $X \in \mathbb{R}^{n \times f}$ is a matrix whose rows correspond to node features.

Graph convolutional networks (GCNs) [27] are a powerful variant of GNNs. For a graph $\mathcal{G} = \{\mathcal{V}, A, X\}$ with a degree matrix $D$, layer $l$ of the GCN architecture can be defined as $H^{(l)} = \sigma(\hat{A} H^{(l-1)} W^{(l)})$ where $\hat{A}$ represents a normalized adjacency matrix, $H^{(l-1)} \in \mathbb{R}^{n \times d_{l-1}}$ represents the node representations in layer *l-1* ($H^{(0)} = X$), $W^{(l)} \in \mathbb{R}^{d_{l-1} \times d_l}$ is a weight matrix, $\sigma$ is an activation function such as ReLU [34], and $H^{(l)} \in \mathbb{R}^{n \times d_l}$ is the updated node embeddings. For undirected graphs where the adjacency is symmetric, $\hat{A} = D^{-\frac{1}{2}}(A + I)D^{-\frac{1}{2}}$ corresponds to a row-and-column normalized adjacency with self-loops, and for directed graphs where the adjacency is not necessarily symmetric, $\hat{A} = D^{-1}(A + I)$ corresponds to a row normalized adjacency matrix with self-loops. Here, $D$ is a (diagonal) degree matrix for $(A + I)$ defined as $D_{ii} = 1 + \sum_j A_{ij}$.

## 4   Proposed method: SLAPS

SLAPS consists of four components: 1) generator, 2) adjacency processor, 3) classifier, and 4) self-supervision. Figure 1 illustrates these components. In the next three subsections, we explain the first three components. Then, we point out a supervision starvation problem for a model based only on these components. Then we describe the self-supervision component as a solution to the supervision starvation problem and the full SLAPS model.

### 4.1   Generator

The generator is a function $\mathsf{G} : \mathbb{R}^{n \times f} \to \mathbb{R}^{n \times n}$ with parameters $\theta_\mathsf{G}$ which takes the node features $X \in \mathbb{R}^{n \times f}$ as input and produces a matrix $\tilde{A} \in \mathbb{R}^{n \times n}$ as output. We consider the following two generators and leave experimenting with more sophisticated graph generators (e.g., [50, 32, 31]) and models with tractable adjacency computations (e.g., [7]) as future work.

**Full parameterization (FP):** For this generator, $\theta_\mathsf{G} \in \mathbb{R}^{n \times n}$ and the generator function is defined as $\tilde{A} = \mathsf{G}_{FP}(X; \theta_\mathsf{G}) = \theta_\mathsf{G}$. That is, the generator ignores the input node features and directly optimizes the adjacency matrix. FP is similar to the generator in LDS [12] except that the generator of LDS treats each element of $\tilde{A}$ as the parameter of a Bernoulli distribution and samples graph structures from these distributions. FP is simple and flexible for learning any adjacency matrix but adds $n^2$ parameters which limits scalability and makes the model susceptible to overfitting.

**MLP-kNN:** Here, $\theta_\mathsf{G}$ corresponds to the weights of a multi-layer perceptron (MLP) and $\tilde{A} = \mathsf{G}_{\mathsf{MLP}}(X; \theta_\mathsf{G}) = \mathsf{kNN}(\mathsf{MLP}(X))$, where $\mathsf{MLP} : \mathbb{R}^{n \times f} \to \mathbb{R}^{n \times f'}$ is an MLP that produces a matrix with updated node representations $X'$; $\mathsf{kNN} : \mathbb{R}^{n \times f'} \to \mathbb{R}^{n \times n}$ produces a sparse matrix. The implementation details for the kNN operation is provided in the supplementary material.

**Initialization and variants of MLP-kNN:** Let $A^{kNN}$ represent an adjacency matrix created by applying a kNN function on the initial node features. One smart initialization for $\theta_\mathsf{G}$ is to initialize it

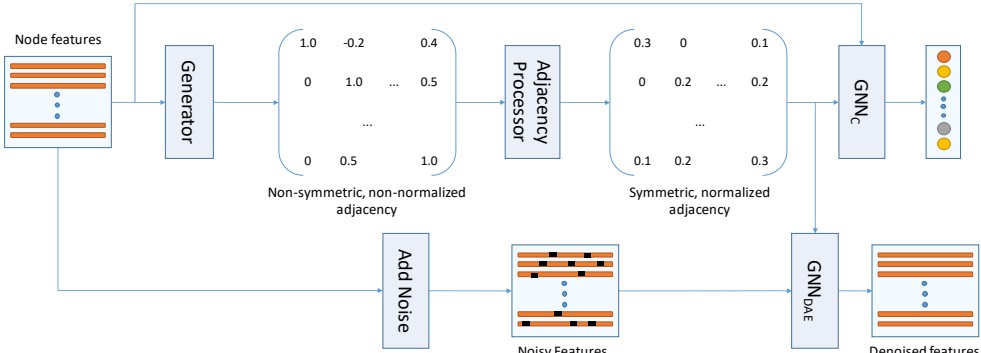

Figure 1: Overview of SLAPS. At the top, a generator receives the node features and produces a non-symmetric, non-normalized adjacency having (possibly) both positive and negative values (Section 4.1). The adjacency processor makes the values positive, symmetrizes and normalizes the adjacency (Section 4.2). The resulting adjacency and the node features go into $\mathsf{GNN_C}$ which predicts the node classes (Section 4.3). At the bottom, some noise is added to the node features. The resulting noisy features and the generated adjacency go into $\mathsf{GNN_{DAE}}$ which then denoises the features (Section 4.5).

in a way that the generator initially generates $\boldsymbol{A}^{kNN}$ (i.e. $\tilde{\boldsymbol{A}} = \boldsymbol{A}^{kNN}$ before training starts). This can be trivially done for the FP generator by initializing $\boldsymbol{\theta}_\mathsf{G}$ to $\boldsymbol{A}^{kNN}$. For MLP-kNN, we consider two variants. In one, hereafter referred to simply as MLP, we keep the input dimension the same throughout the layers. In the other, hereafter referred to as MLP-D, we consider MLPs with diagonal weight matrices (i.e., except the main diagonal, all other parameters in the weight matrices are zero). For both variants, we initialize the weight matrices in $\boldsymbol{\theta}_\mathsf{G}$ with the identity matrix to ensure that the output of the MLP is initially the same as its input and the kNN graph created on these outputs is equivalent to $\boldsymbol{A}^{kNN}$ (alternatively, one may use other MLP variants but pre-train the weights to output $\boldsymbol{A}^{kNN}$ before the main training starts.). MLP-D can be thought of as assigning different weights to different features and then computing node similarities.

## 4.2 Adjacency processor

The output $\tilde{\boldsymbol{A}}$ of the generator may have both positive and negative values, may be non-symmetric and non-normalized. We let $\boldsymbol{A} = \frac{1}{2}\boldsymbol{D}^{-\frac{1}{2}}(\mathsf{P}(\tilde{\boldsymbol{A}}) + \mathsf{P}(\tilde{\boldsymbol{A}})^T)\boldsymbol{D}^{-\frac{1}{2}}$. Here $\mathsf{P}$ is a function with a non-negative range applied element-wise on its input – see supplementary material for details. The sub-expression $\frac{1}{2}(\mathsf{P}(\tilde{\boldsymbol{A}}) + \mathsf{P}(\tilde{\boldsymbol{A}})^T)$ makes the resulting matrix $\mathsf{P}(\tilde{\boldsymbol{A}})$ symmetric. To understand the reason for taking the mean of $\mathsf{P}(\tilde{\boldsymbol{A}})$ and $\mathsf{P}(\tilde{\boldsymbol{A}})^T$, assume $\tilde{\boldsymbol{A}}$ is generated by $\mathsf{G_{MLP}}$. If $v_j$ is among the $k$ most similar nodes to $v_i$ and vice versa, then the strength of the connection between $v_i$ and $v_j$ will remain the same. However, if, say, $v_j$ is among the $k$ most similar nodes to $v_i$ but $v_i$ is not among the top k for $v_j$, then taking the average of the similarities reduces the strength of the connection between $v_i$ and $v_j$. Finally, once we have a symmetric adjacency with non-negative values, we normalize $\frac{1}{2}(\mathsf{P}(\tilde{\boldsymbol{A}}) + \mathsf{P}(\tilde{\boldsymbol{A}})^T)$ by computing its degree matrix $\boldsymbol{D}$ and multiplying it from left and right to $\boldsymbol{D}^{-\frac{1}{2}}$.

## 4.3 Classifier

The classifier is a function $\mathsf{GNN_C} : \mathbb{R}^{n \times f} \times \mathbb{R}^{n \times n} \to \mathbb{R}^{n \times |\mathcal{C}|}$ with parameters $\boldsymbol{\theta}_{\mathsf{GNN_C}}$. It takes the node features $\boldsymbol{X}$ and the generated adjacency $\boldsymbol{A}$ as input and provides for each node the logits for each class. $\mathcal{C}$ corresponds to the classes and $|\mathcal{C}|$ corresponds to the number of classes. We use a two-layer GCN for which $\boldsymbol{\theta}_{\mathsf{GNN_C}} = \{\boldsymbol{W}^{(1)}, \boldsymbol{W}^{(2)}\}$ and define our classifier as $\mathsf{GNN_C}(\boldsymbol{A}, \boldsymbol{X}; \boldsymbol{\theta}_{\mathsf{GNN_C}}) = \boldsymbol{A}\mathsf{ReLU}(\boldsymbol{A}\boldsymbol{X}\boldsymbol{W}^{(1)})\boldsymbol{W}^{(2)}$ but other GNN variants can be used as well (recall that $\boldsymbol{A}$ is normalized). The training loss $\mathcal{L}_C$ for the classification task is computed by taking the softmax of the logits to produce a probability distribution for each node and then computing the cross-entropy loss.

## 4.4 Using only the first three components leads to supervision starvation

One may create a model using only the three components described so far corresponding to the top part of Figure 1. As we will explain here, however, this model may suffer severely from supervision starvation. The same problem also applies to many existing approaches for latent graph learning, as they can be formulated as a combination of variants of these three components.

Consider a scenario during training where two unlabeled nodes $v_i$ and $v_j$ are not directly connected to any labeled nodes according to the generated structure. Then, since a two-layer GCN makes predictions for the nodes based on their two-hop neighbors, the classification loss (i.e. $\mathcal{L}_C$) is not affected by the edge between $v_i$ and $v_j$ and this edge receives no supervision[2]. Figure 2 provides an example of such a scenario. Let us call the edges that do not affect the loss function $\mathcal{L}_C$ (and consequently do not receive supervision) as *starved edges*. These edges are problematic because although they may not affect the training loss, the predictions at the test time depend on these edges and if their values are learned without enough supervision, the model may make poor predictions at the test time. A natural question concerning the extent of the problem caused by such edges is the proportion of starved edges. The following theorem formally establishes the extent of the problem for Erdős-Rényi graphs [10]; in the supplementary, we extend this result to the Barabási–Albert model [1] and scale-free networks [2]. An *Erdős-Rényi* graph with $n$ nodes and $m$ edges is a graph chosen uniformly at random from the collection of all graphs which have $n$ nodes and $m$ edges.

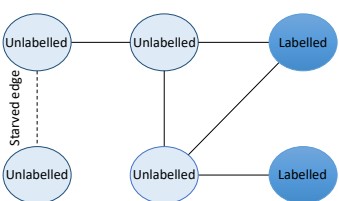

Figure 2: Using a two-layer GCN, the predictions made for the labeled nodes are not affected by the dashed (starved) edge.

**Theorem 1** *Let $\mathcal{G}(n, m)$ be an Erdős-Rényi graph with $n$ nodes and $m$ edges. Assume we have labels for $q$ nodes selected uniformly at random. The probability of an edge being a starved edge with a two-layer GCN is equal to $(1 - \frac{q}{n})(1 - \frac{q}{n-1}) \prod_{i=1}^{2q}(1 - \frac{m-1}{\binom{n}{2}-i})$.*

We defer the proof to the supplementary material. To put the numbers from the theorem in perspective, let us consider three established benchmarks for semi-supervised node classification namely *Cora*, *Citeseer*, and *Pubmed* (the statistics for these datasets can be found in the Appendix). For an Erdős-Rényi graph with similar statistics as the Cora dataset ($n = 2708$, $m = 5429$, $q = 140$), the probability of an edge being a starved edge is $59.4\%$ according to the above theorem. For Citeseer and Pubmed, this number is $75.7\%$ and $96.7\%$ respectively. While Theorem 1 is stated for Erdős-Rényi graphs, the identified problem also applies to natural graphs. For the original structures of Cora, Citeseer, and Pubmed, for example, $48.8\%$, $65.2\%$, and $91.6\%$ of the edges are starved edges.

## 4.5 Self-supervision

One possible solution to the supervision starvation problem is to define a *prior graph structure* and regularize the learned structure toward it. This leads the starved edges toward the prior structure as opposed to neglecting them. The choice of the prior is important as it determines the inductive bias incorporated into the model. We define a prior structure based on the following hypothesis:

**Hypothesis 1** *A graph structure that is suitable for predicting the node features is also suitable for predicting the node labels.*

We first explain why the above hypothesis is reasonable for an extreme case that is easy to understand and then extend the explanation to the general case. Consider an extreme scenario where one of the node features is the same as the node labels. A graph structure that is suitable for predicting this feature exhibits homophily for it. Because of the equivalence between this feature and the labels, the graph structure also exhibits homophily for the labels, so it is also suitable for predicting the labels. In the general (non-extreme) case, there may not be a single feature that is equivalent to the labels but a subset of the features may be highly predictive of the labels. A graph structure that is suitable for predicting this subset exhibits homophily for the features in the subset. Because this subset is highly

---

[2]While using more layers may somewhat alleviate this problem, deeper GCNs typically produce inferior results, e.g., due to oversmoothing [29, 35] – see the supplementary material for empirical evidence.

predictive of the labels, the structure also exhibits a high degree of homophily for the labels, so it is also suitable for predicting the node labels.

Next, we explain how to design a suitable graph structure for predicting the features and how to regularize toward it. One could design such a structure manually (e.g., by handcrafting a graph that connects nodes based on the collective homophily between their individual features) and then penalize the difference between this prior graph and the learned graph. Alternatively, in this paper, we take a learning-based approach based on self-supervision where we not only use the learned graph structure for the classification task, but also for denoising the node features. The self-supervised task encourages the model to learn a structure that is suitable for predicting the node features. We describe this approach below and provide comparisons to the manual approach in the supplementary material.

Our self-supervised task is based on denoising autoencoders [46]. Let $\mathsf{GNN}_{\mathsf{DAE}} : \mathbb{R}^{n \times f} \times \mathbb{R}^{n \times n} \to \mathbb{R}^{n \times f}$ be a GNN with parameters $\boldsymbol{\theta}_{\mathsf{GNN}_{\mathsf{DAE}}}$ that takes node features and a generated adjacency as input and provides updated node features with the same dimension as output. We train $\mathsf{GNN}_{\mathsf{DAE}}$ such that it receives a noisy version $\tilde{\boldsymbol{X}}$ of the features $\boldsymbol{X}$ as input and produces the denoised features $\boldsymbol{X}$ as output. Let $idx$ represent the indices corresponding to the elements of $\boldsymbol{X}$ to which we have added noise, and $\boldsymbol{X}_{idx}$ represent the values at these indices. During training, we minimize:

$$\mathcal{L}_{DAE} = \mathsf{L}(\boldsymbol{X}_{idx}, \mathsf{GNN}_{\mathsf{DAE}}(\tilde{\boldsymbol{X}}, \boldsymbol{A}; \boldsymbol{\theta}_{\mathsf{GNN}_{\mathsf{DAE}}})_{idx}) \tag{1}$$

where $\boldsymbol{A}$ is the generated adjacency matrix and $\mathsf{L}$ is a loss function. For datasets where the features consist of binary vectors, $idx$ consists of $r$ percent of the indices of $\boldsymbol{X}$ whose values are 1 and $r\eta$ percent of the indices whose values are 0, both selected uniformly at random in each epoch. Both $r$ and $\eta$ (corresponding to the negative ratio) are hyperparameters. In this case, we add noise by setting the 1s in the selected mask to 0s and $\mathsf{L}$ is the binary cross-entropy loss. For datasets where the input features are continuous numbers, $idx$ consists of $r$ percent of the indices of $\boldsymbol{X}$ selected uniformly at random in each epoch. We add noise by either replacing the values at $idx$ with 0 or by adding independent Gaussian noises to each of the features. In this case, $\mathsf{L}$ is the mean-squared error loss.

Note that the self-supervised task in equation 1 is generic and can be added to different GNNs as well as latent graph learning models. It can be also combined with other techniques in the literature that encourage learning more homophilous structures or increase the amount of supervision. In our experiments, we test the combination of our self-supervised task with two such techniques namely *self-training* [29] and *AdaEdge* [5]. Self-training helps the model "see" more labeled nodes and AdaEdge helps iteratively create graph structure with higher degrees of homophily. We refer the reader to the supplementary material for descriptions of self-training and AdaEdge.

### 4.6 SLAPS

Our final model is trained to minimize $\mathcal{L} = \mathcal{L}_C + \lambda \mathcal{L}_{DAE}$ where $\mathcal{L}_C$ is the classification loss, $\mathcal{L}_{DAE}$ is the denoising autoencoder loss (see Equation 1), and $\lambda$ is a hyperparameter controlling the relative importance of the two losses.

## 5 Experiments

In this section, we report our key results. More empirical comparisons, experimental analyses, and ablation studies are presented in the supplementary material.

**Baselines:** We compare our proposal to several baselines with different properties. The first baseline is a multi-layer perceptron (MLP) which does not take the graph structure into account. We also compare against MLP-GAM* [42] which learns a fully connected graph structure and uses this structure to supplement the loss function of the MLP toward predicting similar labels for neighboring nodes. Our third baseline is label propagation (LP) [59], a well-known model for semi-supervised learning. Similar to [12], we also consider a baseline named *kNN-GCN* where we create a kNN graph based on the node feature similarities and feed this graph to a GCN; the graph structure remains fixed in this approach. We also compare with prominent existing latent graph learning models including LDS [12], GRCN [52], DGCNN [47], and IDGL [6]. In [6], another variant named IDGL-ANCH is also proposed that reduces time complexity through anchor-based approximation [33]. We compare against the base IDGL model because it does not sacrifice accuracy for time complexity, and because anchor-based approximation is model-agnostic and could be combined with other models too. We

Table 1: Results of SLAPS and the baselines on established node classification benchmarks. †
indicates results have been taken from Franceschi et al. [12]. ‡ indicates results have been taken from
Stretcu et al. [42]. Bold and underlined values indicate best and second-best mean performances
respectively. *OOM* indicates out of memory. *OOT* indicates out of time (we allowed 24h for each
run). *NA* indicates not applicable.

| Model | Cora | Citeseer | Cora390 | Citeseer370 | Pubmed | ogbn-arxiv |
|---|---|---|---|---|---|---|
| MLP | $56.1 \pm 1.6^{\dagger}$ | $56.7 \pm 1.7^{\dagger}$ | $65.8 \pm 0.4$ | $67.1 \pm 0.5$ | $71.4 \pm 0.0$ | $\underline{54.7 \pm 0.1}$ |
| MLP-GAM* | $70.7^{\ddagger}$ | $70.3^{\ddagger}$ | − | − | $71.9^{\ddagger}$ | − |
| LP | $37.6 \pm 0.0$ | $23.2 \pm 0.0$ | $36.2 \pm 0.0$ | $29.1 \pm 0.0$ | $41.3 \pm 0.0$ | OOM |
| kNN-GCN | $66.5 \pm 0.4^{\dagger}$ | $68.3 \pm 1.3^{\dagger}$ | $72.5 \pm 0.5$ | $71.8 \pm 0.8$ | $70.4 \pm 0.4$ | $49.1 \pm 0.3$ |
| LDS | − | − | $71.5 \pm 0.8^{\dagger}$ | $71.5 \pm 1.1^{\dagger}$ | OOM | OOM |
| GRCN | $67.4 \pm 0.3$ | $67.3 \pm 0.8$ | $71.3 \pm 0.9$ | $70.9 \pm 0.7$ | $67.3 \pm 0.3$ | OOM |
| DGCNN | $56.5 \pm 1.2$ | $55.1 \pm 1.4$ | $67.3 \pm 0.7$ | $66.6 \pm 0.8$ | $70.1 \pm 1.3$ | OOM |
| IDGL | $70.9 \pm 0.6$ | $68.2 \pm 0.6$ | $73.4 \pm 0.5$ | $72.7 \pm 0.4$ | $72.3 \pm 0.4$ | OOM |
| kNN-GCN + AdaEdge | $67.7 \pm 1.0$ | $68.8 \pm 1.2$ | $72.2 \pm 0.4$ | $71.8 \pm 0.6$ | OOT | OOT |
| kNN-GCN + self-training | $67.3 \pm 0.3$ | $69.8 \pm 1.0$ | $71.1 \pm 0.3$ | $72.4 \pm 0.2$ | $72.7 \pm 0.1$ | NA |
| SLAPS (FP) | $72.4 \pm 0.4$ | $70.7 \pm 0.4$ | $\mathbf{76.6 \pm 0.4}$ | $73.1 \pm 0.6$ | OOM | OOM |
| SLAPS (MLP) | $72.8 \pm 0.8$ | $70.5 \pm 1.1$ | $75.3 \pm 1.0$ | $73.0 \pm 0.9$ | $\mathbf{74.4 \pm 0.6}$ | $\mathbf{56.6 \pm 0.1}$ |
| SLAPS (MLP-D) | $\underline{73.4 \pm 0.3}$ | $72.6 \pm 0.6$ | $75.1 \pm 0.5$ | $\mathbf{73.9 \pm 0.4}$ | $73.1 \pm 0.7$ | $52.9 \pm 0.1$ |
| SLAPS (MLP) + AdaEdge | $72.8 \pm 0.7$ | $70.6 \pm 1.5$ | $75.2 \pm 0.6$ | $72.6 \pm 1.4$ | OOT | OOT |
| SLAPS (MLP) + self-training | $\mathbf{74.2 \pm 0.5}$ | $\mathbf{73.1 \pm 1.0}$ | $\underline{75.5 \pm 0.7}$ | $\underline{73.3 \pm 0.6}$ | $74.3 \pm 1.4$ | NA |

feed a kNN graph to the models requiring an initial graph structure. We also explore how adding
self-training and AdaEdge impact the performance of kNN-GCN as well as SLAPS.

**Datasets:** We use three established benchmarks in the GNN literature namely Cora, Citeseer, and
Pubmed [41] as well as the *ogbn-arxiv* dataset [16] that is orders of magnitude larger than the
other three datasets and is more challenging due to the more realistic split of the data into train,
validation, and test sets. For these datasets, we only feed the node features to the models and not their
original graph structure. Following [12, 6], we also experiment with several classification (non-graph)
datasets available in scikit-learn [36] including Wine, Cancer, Digits, and 20News. Furthermore,
following [20], we also provide results on MNIST [28]. The dataset statistics can be found in the
supplementary. For Cora and Citeseer, the LDS model uses the train data for learning the parameters
of the classification GCN, half of the validation for learning the parameters of the adjacency matrix
(in their bi-level optimization setup, these are considered as hyperparameters), and the other half of
the validation set for early stopping and tuning the other hyperparameters. Besides experimenting
with the original setups of these two datasets, we also consider a setup that is closer to that of LDS:
we use the train set and half of the validation set for training and the other half of validation for
early stopping and hyperparameter tuning. We name the modified versions Cora390 and Citeseer370
respectively where the number proceeding the dataset name shows the number of labels from which
gradients are computed. We follow a similar procedure for the scikit-learn datasets.

**Implementation:** We defer the implementation details and the best hyperparameter settings for our
model on all the datasets to the supplementary material.

## 5.1 Comparative results

The results of SLAPS and the baselines on our benchmarks are reported in Tables 1 and 2. We start by
analyzing the results in Table 1 first. Starting with the baselines, we see that learning a fully connected
graph in MLP-GAM* makes it outperform MLP. kNN-GCN significantly outperforms MLP on Cora
and Citeseer but underperforms on Pubmed and ogbn-arxiv. Furthermore, both self-training and
AdaEdge improve the performance of kNN-GCN. This shows the importance of the similarity metric
and the graph structure that is fed into GCN; a low-quality structure can harm model performance.
LDS outperforms MLP but the fully parameterized adjacency matrix of LDS results in memory issues
for Pubmed and ogbn-arxiv. As for GRCN, it was shown in the original paper that GRCN can revise a
good initial adjacency matrix and provide a substantial boost in performance. However, as evidenced
by the results, if the initial graph structure is somewhat poor, GRCN's performance becomes on par
with kNN-GCN. IDGL is the best performing baseline.

Table 2: Results on classification datasets. † indicates results have been taken from Franceschi et al. [12]. Bold and underlined values indicate best and second-best mean performances respectively.

| Model | Wine | Cancer | Digits | 20news |
|---|---|---|---|---|
| MLP | $96.1 \pm 1.0$ | $95.3 \pm 0.9$ | $81.9 \pm 1.0$ | $30.4 \pm 0.1$ |
| kNN-GCN | $93.5 \pm 0.7$ | $95.3 \pm 0.4$ | $\mathbf{95.4 \pm 0.4}$ | $46.3 \pm 0.3$ |
| LDS | $\mathbf{97.3 \pm 0.4}^{\dagger}$ | $94.4 \pm 1.9^{\dagger}$ | $92.5 \pm 0.7^{\dagger}$ | $46.4 \pm 1.6^{\dagger}$ |
| IDGL | $97.0 \pm 0.7$ | $94.2 \pm 2.3$ | $92.5 \pm 1.3$ | $48.5 \pm 0.6$ |
| SLAPS (FP) | $96.6 \pm 0.4$ | $94.6 \pm 0.3$ | $\underline{94.4 \pm 0.7}$ | $44.4 \pm 0.8$ |
| SLAPS (MLP) | $96.3 \pm 1.0$ | $\underline{96.0 \pm 0.8}$ | $92.5 \pm 0.7$ | $\mathbf{50.4 \pm 0.7}$ |
| SLAPS (MLP-D) | $96.5 \pm 0.8$ | $\mathbf{96.6 \pm 0.2}$ | $94.2 \pm 0.1$ | $\underline{49.8 \pm 0.9}$ |

In addition to the aforementioned baselines, we also experimented with GCN, GAT, and Transformer (encoder only) architectures applied on fully connected graphs. GCN always learned to predict the majority class. This is because after one fully connected GCN layer, all nodes will have the same embedding and become indistinguishable. GAT also showed similar behavior. We believe this is because the attention weights are (almost) random at the beginning (due to random initialization of the model parameters) resulting in nodes becoming indistinguishable and GAT cannot escape from that state. The skip connections of Transformer helped avoid the problem observed for GCN and GAT and we were able to achieve better results ($\sim 40\%$ accuracy on Cora). However, we observed severe overfitting, even with small models and with high dropout probabilities.

SLAPS consistently outperforms the baselines in some cases by large margins. Among the generators, the winner is dataset-dependent with MLP-D mostly outperforming MLP on datasets with many features and MLP outperforming on datasets with small numbers of features. Using the software that was publicly released by the authors, the baselines that learn a graph structure fail on ogbn-arxiv; our implementation, on the other hand, scales to such large graphs[3]. Adding self-training helps further improve the results of SLAPS. Adding AdaEdge, however, does not seem effective, probably because the graph structure learned by SLAPS already exhibits a high degree of homophily (see Section 5.4).

In Table 2, we only compared SLAPS with the best performing baselines from Table 1 (kNN-GCN, LDS and IDGL). We also included an MLP baseline for comparison. On three out of four datasets, SLAPS outperforms the LDS and IDGL baselines. For the Digits dataset, interestingly kNN-GCN outperforms the learning-based models. This could be because the initial kNN structure for this dataset is already a good structure. Among the datasets on which we can train SLAPS with the FP generator, 20news has the largest number of nodes (9,607 nodes). On this dataset, we observed that an FP generator suffers from overfitting and produces weaker results compared to other generators due to its large number of parameters.

Jiang et al. [21] show that learning a latent graph structure of the input examples can help with semi-supervised image classification. In particular, they create three versions of the MNIST dataset each consisting of a randomly selected subset with 10,000 examples in total. The first version contains 1000 labels for training, the second contains 2000, and the third version contains 3000 labels for training. All three variants use an extra 1000 labels for validation. The other examples are used as test examples. Here, we conduct an experiment to measure the performance of SLAPS on these variants of the MNIST dataset. We compare against GLCN [21] as well as the baselines in the GLCN paper including manifold regularization [3], label propagation, deep walk [37], graph convolutional networks (GCN), and graph attention networks (GAT).

Table 3: Results on the MNIST dataset. Bold values indicate best mean performances. Underlined values indicate second best mean performance. All the results for baseline have been taken from [20].

| Model | MNIST1000 | MNIST2000 | MNIST3000 |
|---|---|---|---|
| ManiReg | $92.74 \pm 0.3$ | $93.96 \pm 0.2$ | $94.62 \pm 0.2$ |
| LP | $79.28 \pm 0.9$ | $81.91 \pm 0.8$ | $83.45 \pm 0.5$ |
| DeepWalk | $\underline{94.55 \pm 0.3}$ | $95.04 \pm 0.3$ | $95.34 \pm 0.3$ |
| GCN | $90.59 \pm 0.3$ | $90.91 \pm 0.2$ | $91.01 \pm 0.2$ |
| GAT | $92.11 \pm 0.4$ | $92.64 \pm 0.3$ | $92.81 \pm 0.3$ |
| GLCN | $94.28 \pm 0.3$ | $\underline{95.09 \pm 0.2}$ | $\underline{95.46 \pm 0.2}$ |
| SLAPS | $\mathbf{94.66 \pm 0.2}$ | $\mathbf{95.35 \pm 0.1}$ | $\mathbf{95.54 \pm 0.0}$ |

The results are reported in Table 3. From the results, it can be viewed that SLAPS outperforms GLCN and all the other baselines on the 3 variants. Compared to GLCN, on the three variants SLAPS

---

[3]We note that IDGL-ANCH also scales to ogbn-arxiv.

reduces the error by $7\%$, $5\%$, and $2\%$ respectively, showing that SLAPS can be more effective when the labeled set is small and providing more empirical evidence for Theorem 1.

## 5.2 The effectiveness of self-supervision

**Learning a structure only using self-supervision:** To provide more insight into the value provided by the self-supervision task and the generalizability of the adjacency learned through this task, we conduct experiments with a variant of SLAPS named $SLAPS_{2s}$ that is trained in two stages. We first train the GNN$_{DAE}$ model by minimizing $\mathcal{L}_{DAE}$ described in in Equation 1. Recall that $\mathcal{L}_{DAE}$ depends on the parameters $\boldsymbol{\theta}_G$ of the generator and the parameters $\boldsymbol{\theta}_{GNN_{DAE}}$ of the denoising autoencoder. After every $t$ epochs of training, we fix the adjacency matrix, train a classifier with the fixed adjacency matrix, and measure classification accuracy on the validation set. We select the epoch that produces the adjacency providing the best validation accuracy for the classifier. Note that in $SLAPS_{2s}$, the adjacency matrix only receives gradients from the self-supervised task in Equation 1.

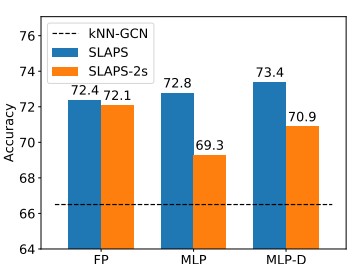

Figure 3: SLAPS vs SLAPS$_{2s}$ on Cora with different generators.

Figure 3 shows the performance of SLAPS and SLAPS$_{2s}$ on Cora and compares them with kNN-GCN. Although SLAPS$_{2s}$ does not use the node labels in learning an adjacency matrix, it outperforms kNN-GCN ($8.4\%$ improvement when using an FP generator). With an FP generator, SLAPS$_{2s}$ even achieves competitive performance with SLAPS; this is mainly because FP does not leverage the supervision provided by GCN$_C$ toward learning generalizable patterns that can be used for nodes other than those in the training set. These results corroborate the effectiveness of the self-supervision task for learning an adjacency matrix. Besides, the results show that learning the adjacency using both self-supervision and the task-specific node labels results in higher predictive accuracy.

**The value of $\lambda$:** Figure 4 shows the performance of SLAPS[4] on Cora and Citeseer with different values of $\lambda$. When $\lambda = 0$, corresponding to removing self-supervision, the model performance is somewhat poor. As soon as $\lambda$ becomes positive, both models see a large boost in performance showing that self-supervision is crucial to the high performance of SLAPS. Increasing $\lambda$ further provides larger boosts until it becomes so large that the self-supervision loss dominates the classification loss and the performance deteriorates. Note that with $\lambda = 0$, SLAPS with the MLP generator becomes a variant of the model proposed in [8], but with a different similarity function.

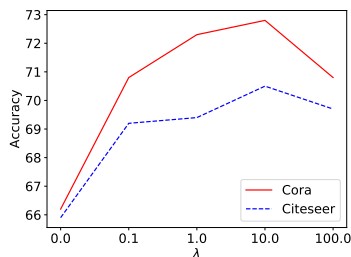

Figure 4: The performance of SLAPS with MLP graph generator as a function of $\lambda$.

**Is self-supervision actually solving the supervision starvation problem?** In Fig 4, we showed that self-supervision is key to the high performance of SLAPS. Here, we examine if this is because self-supervision indeed addresses the supervision starvation problem. For this purpose, we compare SLAPS with and without self-supervision on two groups of test nodes on Cora: 1) those that are not connected to any labeled nodes after training, and 2) those that are connected to at least one labeled node after training. The nodes in group one have a high chance of having starved edges. We observed that adding self-supervision provides $38.0\%$ improvement for the first group and only $8.9\%$ improvement for the latter. Since self-supervision mainly helps with nodes in group 1, this provides evidence that self-supervision is an effective solution to the supervision starvation problem.

**The effect of the training set size:** According to Theorem 1, a smaller $q$ (corresponding to the training set size) results in more starved edges in each epoch. To explore the effect of self-supervision as a function of $q$, we compared SLAPS with and without supervision on Cora and Citeseer while reducing the number of labeled nodes per class from 20 to 5. We used the FP generator for this experiment. With 5 labeled nodes per class, adding self-supervision provides $16.7\%$ and $22.0\%$

---

[4]The generator used in this experiment is MLP; other generators produced similar results.

improvements on Cora and Citeseer respectively, which is substantially higher than the corresponding numbers when using 20 labeled nodes per class (10.0% and 7.0% respectively). This provides empirical evidence for Theorem 1. Note that the results on Cora390 and Citeseer 370 datasets provide evidence that the self-supervised task is effective even when the label rate is high.

## 5.3 Experiments with noisy graphs

The performance of GNNs highly depends on the quality of the input graph structure and deteriorates when the graph structure is noisy [see 61, 9, 11]. Here, we verify whether self-supervision is also helpful when a noisy structure is provided as input. Toward this goal, we experiment with Cora and Citeseer and provide noisy versions of the input graph as input. The provided noisy graph structure is used only for initialization; it is then further optimized by SLAPS. We perturb the graph structure by replacing $\rho$ percent of the edges in the original structure (selected uniformly at random) with random edges. Figure 5 shows the performance of SLAPS with and without self-supervision ($\lambda = 0$ corresponds to no supervision). We also report the results of vanilla GCN on these perturbed graphs for comparison. It can be viewed that self-supervision consistently provides a boost in performance especially for higher values of $\rho$.

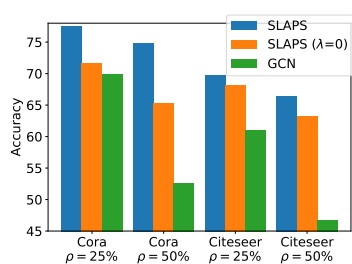

Figure 5: Performance comparison when noisy graphs are provided as input ($\rho$ indicates the percentage of perturbations).

## 5.4 Analyses of the learned adjacency

**Noisy graphs:** Following the experiment in Section 5.3, we compared the learned and original structures by measuring the number of random edges added during perturbation but removed by the model and the number of edges removed during the perturbation but recovered by the model. For Cora, SLAPS removed 76.2% and 70.4% of the noisy edges and recovered 58.3% and 44.5% of the removed edges for $\rho = 25\%$ and $\rho = 50\%$ respectively while SLAPS with $\lambda = 0$ only removed 62.8% and 54.9% of the noisy edges and recovered 51.4% and 35.8% of the removed edges. This provides evidence on self-supervision being helpful for structure learning.

**Homophily:** As explained earlier, a properly learned graph for semi-supervised classification with GNNs exhibits high homophily. To verify the quality of the learned adjacency with respect to homophily, for every pair of nodes in the test set, we compute the odds of the two nodes sharing the same label as a function of the normalized weight of the edge connecting them. Figure 6 represents the odds for different weight intervals (recall that $A$ is row and column normalized). For both Cora and Citeseer, nodes' connected with higher edge weights are more likely to share the same label compared to nodes with lower or zero edge weights. Specifically, when $A_{ij} \geq 0.1$, $v_i$ and $v_j$ are almost 2.5 and 2.0 times more likely to share the same label on Cora and Citeseer respectively.

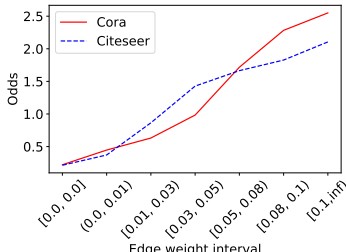

Figure 6: The odds of two nodes in the test set sharing the same label as a function of the edge weights learned by SLAPS.

## 6 Conclusion

We proposed SLAPS: a model for learning the parameters of a graph neural network and a graph structure of the nodes connectivities simultaneously from data. We identified a supervision starvation problem that emerges for graph structure learning, especially when training data is scarce. We proposed a solution to the supervision starvation problem by supplementing the training objective with a well-motivated self-supervised task. We showed the effectiveness of our model through a comprehensive set of experiments and analyses.

## 7 Funding Transparency Statement

This work was fully funded by Borealis AI.

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
