# SLAPS: Self-Supervision Improves Structure Learning for Graph Neural Networks – Supplementary Material

**Bahare Fatemi**[*]
University of British Columbia
bfatemi@cs.ubc.ca

**Layla El Asri**
Borealis AI
layla.elasri@borealisai.com

**Seyed Mehran Kazemi**[*]
Google Research
mehrankazemi@google.com

## A   More Experiments and Analyses

**Importance of $k$ in kNN:** Figure 1 shows the performance of SLAPS on Cora for three graph generators as a function of $k$ in kNN. For all three cases, the value of $k$ plays a major role in model performance. The FP generator is the least sensitive because, in FP, $k$ only affects the initialization of the adjacency matrix but then the model can change the number of neighbors of each node. For MLP and MLP-D, however, the number of neighbors of each node remains close to $k$ (but not necessarily equal as the adjacency processor can add or remove some edges) and the two generators become more sensitive to $k$. For larger values of $k$, the extra flexibility of the MLP generator enables removing some of the unwanted edges through the function P or reducing the weights of the unwanted edges resulting in MLP being less sensitive to large values of $k$ compared to MLP-D.

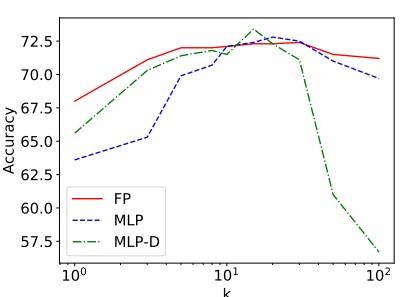

Figure 1: The performance of SLAPS on Cora as a function of $k$ in kNN.

**Increasing the number of layers:** In the main text, we described how some edges may receive no supervision during latent graph learning. We pointed out that while increasing the number of layers of the GCN may alleviate the problem to some extent, deeper GCNs typically provide inferior results due to issues such as oversmoothing [see, e.g., 9, 10]. We empirically tested deeper GCNs for latent graph learning to see if simply using more layers can obviate the need for the proposed self-supervision. Specifically, we tested SLAPS without self-supervision (i.e. $\lambda = 0$) with 2, 4, and 6 layers on Cora. We also added residual connections that have been shown to help train deeper GCNs [8]. The accuracies for 2, 4, and 6-layer models are 66.2%, 67.1%, and 55.8% respectively. It can be viewed that increasing the number of layers from 2 to 4 provides an improvement. This might be because the benefit provided by a 4-layer model in terms of alleviating the starved edge problem outweighs the increase in oversmoothing. However, when the number of layers increases to 6, the oversmoothing problem outweighs and the performance drops significantly. Further increasing the number of layers resulted in even lower accuracies.

---

[*]Work was done when authors were at Borealis AI.

35th Conference on Neural Information Processing Systems (NeurIPS 2021).

**Symmetrization:** In the adjacency processor, we used the following equation:

$$\boldsymbol{A} = \boldsymbol{D}^{-\frac{1}{2}} \Big( \frac{\mathsf{P}(\tilde{\boldsymbol{A}}) + \mathsf{P}(\tilde{\boldsymbol{A}})^T}{2} \Big) \boldsymbol{D}^{-\frac{1}{2}}$$

which symmetrized the adjacency matrix by taking the average of $\mathsf{P}(\tilde{\boldsymbol{A}})$ and $\mathsf{P}(\tilde{\boldsymbol{A}})^T$. Here we also consider two other choices: 1) $\max(\mathsf{P}(\tilde{\boldsymbol{A}}), \mathsf{P}(\tilde{\boldsymbol{A}})^T)$, and 2) not symmetrizing the adjacency (i.e. using $\mathsf{P}(\tilde{\boldsymbol{A}})$). Figure 2 compares these three choices on Cora and Citeseer with an MLP generator (other generators produced similar results). On both datasets, symmetrizing the adjacency provides a performance boost. Compared to mean symmetrization, max symmetrization performs slightly worse. This may be because max symmetrization does not distinguish between the case where both $v_i$ and $v_j$ are among the $k$ most similar nodes of each other and the case where only one of them is among the $k$ most similar nodes of the other.

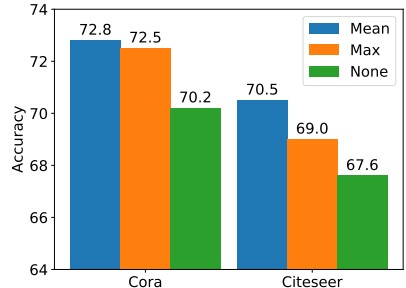

Figure 2: The performance of SLAPS on Cora and Citeseer with different adjacency symmetrizations.

**Fixing a prior graph manually instead of using self-supervision:** In the main text, we validated Hypothesis 1 by adding a self-supervised task to encourage learning a graph structure that is appropriate for predicting the node features, and showing in our experiments how this additional task helps improve the results. Here, we provide more evidence for the validity of Hypothesis 1 by showing that we can obtain good results even when regularizing the learned graph structure toward a manually fixed structure that is appropriate for predicting the node features.

Toward this goal, we experimented with Cora and Citeseer and created a cosine similarity graph as our prior graph $\boldsymbol{A}^{prior}$ where the edge weights correspond to the cosine similarity of the nodes. We sparsified $\boldsymbol{A}^{prior}$ by connecting each node only to the $k$ most similar nodes. Then, we added a term $\lambda ||\boldsymbol{A} - \boldsymbol{A}^{prior}||_F$ to the loss function where $\lambda$ is a hyperparameter, $\boldsymbol{A}$ is the learned graph structure (i.e. the output of the graph generator), and $||.||_F$ shows the Frobenius norm. Note that $\boldsymbol{A}^{prior}$ exhibits homophily with respect to the node features because the node features in Cora and Citeseer are binary, so two nodes that share the same values for more features have a higher similarity and are more likely to be connected.

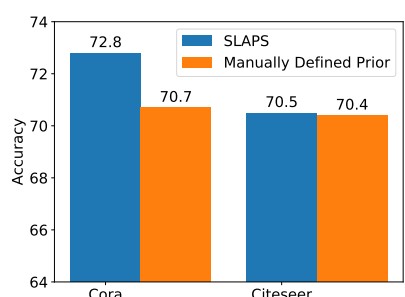

Figure 3: The performance of SLAPS and regularization toward a manually defined prior structure on Cora and Citeseer when using the MLP generator.

The results can be viewed in Figure 3. According to the results, we can see that regularizing toward a manually designed $\boldsymbol{A}^{prior}$ also provides good results but falls short of SLAPS with self-supervision. The superiority of the self-supervised approach compared to the manual design could be due to two reasons.

- Some of the node features may be redundant (e.g., they may be derived from other features) or highly correlated. These features can negatively affect the similarity computations for the prior graph in $\boldsymbol{A}^{prior}$. As an example, consider three nodes with seven binary features $[0, 0, 0, 1, 1, 1, 1]$, $[0, 0, 0, 0, 0, 0, 0]$ and $[1, 1, 1, 1, 1, 1, 1]$ respectively and assume the last two features for each node are always equivalent and are computed based on a *logical and* of the 4th and 5th features[2]. Without these two features, the first node is more similar to the second than the third node, but when considering these derived features, it becomes more similar to the third node. This change in node similarities affects the construction of $\boldsymbol{A}^{prior}$ which can deteriorate the overall performance of the model. The version of SLAPS with the self-supervised task, on the other hand, is not affected by this problem as much because the

---

[2]For the first node in the example, the 4th and 5th features are both $1$ so their logical and is also $1$ and so the last two features for this node are both $1$. The computation for the other two nodes is similar.

model can learn to predict the derived node features based on other features and without heavily relying on the graph structure.

- While many graph structures may be appropriate for predicting the node features, in the manual approach we only regularize toward one particular such structure. Using the self-supervised task, however, SLAPS can learn any of those structures; ideally, it learns the one that is more suited for the downstream task due to the extra supervision coming from the downstream task.

## B  Implementation Details

We implemented our model in PyTorch [11], used deep graph library (DGL) [14] for the sparse operations, and used Adam [6] as the optimizer. We performed early stopping and hyperparameter tuning based on the accuracy on the validation set for all datasets except Wine and Cancer. For these two datasets, the validation accuracy reached 100 percent with many hyperparameter settings, making it difficult to select the best set of hyperparameters. Instead, we used the validation cross-entropy loss for these two datasets.

We fixed the maximum number of epochs to 2000. We use two-layer GCNs for both $GNN_C$ and $GNN_{DAE}$ as well as for baselines and two-layer MLPs throughout the paper (for experiments on ogbn-arxiv, although the original paper uses models with three layers and with batch normalization after each layer, to be consistent with our other experiments we used two layers and removed the normalization). We used two learning rates, one for $GCN_C$ as $lr_C$ and one for the other parameters of the models as $lr_{DAE}$. We tuned the two learning rates from the set $\{0.01, 0.001\}$. We added dropout layers with dropout probabilities of $0.5$ after the first layer of the GNNs. We also added dropout to the adjacency matrix for both $GNN_C$ and $GNN_{DAE}$ as $dropout_C$ $dropout_{DAE}$ respectively and tuned the values from the set $\{0.25, 0.5\}$. We set the hidden dimension of $GNN_C$ to 32 for all datasets except for ogbn-arxiv for which we set it to 256. We used cosine similarity for building the kNN graphs and tuned the value of $k$ from the set $\{10, 15, 20, 30\}$. We tuned $\lambda$ ($\lambda$ controls the relative importance of the two losses) from the set $\{0.1, 1, 10, 100, 500\}$. We tuned $r$ and $\eta$ from the sets $\{1, 5, 10\}$ and $\{1, 5\}$ respectively. The best set of hyperparameters for each dataset chosen on the validation set is in table 1. The code of our experiments will be available upon acceptance of the paper.

For GRCN [16], DGCNN [15], and IDGL [2], we used the code released by the authors and tuned the hyperparameters as suggested in the original papers. The results of LDS [4] are directly taken from the original paper. For LP [18], we used scikit-learn python package [12].

All the results for our model and the baselines are averaged over 10 runs. We report the mean and standard deviation. We ran all the experiments on a single GPU (NVIDIA GeForce GTX 1080 Ti).

**Self-training and AdaEdge:** We combined SLAPS (and kNN-GCN) with two techniques from the literature namely *self-training* and *AdaEdge*. For completeness sake, we provide a brief description of these approaches and refer the reader to the original papers for detailed descriptions.

For self-training, we first trained a model using the existing labels in the training set. Then we used this model to make predictions for the unlabeled nodes that were not in the train, validation, or test sets. We considered the label predictions for the top $\zeta$ most confident unlabeled nodes as ground truth labels and added them to the training labels. Finally, we trained a model from scratch on the expanded set of labels. Here, $\zeta$ is a hyperparameter. We tuned its value from the set $\{50, 100, 200, 300, 400, 500\}$.

For AdaEdge, in the case of kNN-GCN, we first trained a kNN-GCN model. Then we changed the structure of the graph from the kNN graph to a new graph by following these steps: 1) add edges between nodes with the same class predictions if both prediction confidences surpass a threshold, 2) remove edge between nodes with different class predictions if both prediction confidences surpass a threshold. Then, we trained a GCN model on the new structure and repeated the aforementioned steps to generate a new structure. We did this iteratively until generating a new structure did not provide a boost in performance on the validation set. For SLAPS, we followed a similar approach except that the initial model was a SLAPS model instead of a kNN-GCN model.

**kNN Implementation:** For our MLP generator, we used a kNN operation to sparsify the generated graph. Here, we explain how we implemented the kNN operation to avoid blocking the gradient flow. Let $M \in \mathbb{R}^{n \times n}$ with $M_{ij} = 1$ if $v_j$ is among the top $k$ similar nodes to $v_i$ and 0 otherwise, and let $S \in \mathbb{R}^{n \times n}$ with $S_{ij} = \mathsf{Sim}(X'_i, X'_j)$ for some differentiable similarity function Sim (we

Table 1: Best set of hyperparameters for different datasets chosen on validation set.

| Dataset | Generator | $lr_C$ | $lr_{DAE}$ | $dropout_c$ | $dropout_{DAE}$ | $k$ | $\lambda$ | $r$ | $\eta$ |
|---|---|---|---|---|---|---|---|---|---|
| Cora | FP | 0.001 | 0.01 | 0.5 | 0.25 | 30 | 10 | 10 | 5 |
| Cora | MLP | 0.01 | 0.001 | 0.25 | 0.5 | 20 | 10 | 10 | 5 |
| Cora | MLP-D | 0.01 | 0.001 | 0.25 | 0.5 | 15 | 10 | 10 | 5 |
| Citeseer | FP | 0.01 | 0.01 | 0.5 | 0.5 | 30 | 1 | 10 | 1 |
| Citeseer | MLP | 0.01 | 0.001 | 0.25 | 0.5 | 30 | 10 | 10 | 5 |
| Citeseer | MLP-D | 0.001 | 0.01 | 0.5 | 0.5 | 20 | 10 | 10 | 5 |
| Cora390 | FP | 0.01 | 0.01 | 0.25 | 0.5 | 20 | 100 | 10 | 5 |
| Cora390 | MLP | 0.01 | 0.001 | 0.25 | 0.5 | 20 | 10 | 10 | 5 |
| Cora390 | MLP-D | 0.001 | 0.001 | 0.25 | 0.5 | 20 | 10 | 10 | 5 |
| Citeseer370 | FP | 0.01 | 0.01 | 0.5 | 0.5 | 30 | 1 | 10 | 1 |
| Citeseer370 | MLP | 0.01 | 0.001 | 0.25 | 0.5 | 30 | 10 | 10 | 5 |
| Citeseer370 | MLP-D | 0.01 | 0.01 | 0.25 | 0.5 | 20 | 10 | 10 | 5 |
| Pubmed | MLP | 0.01 | 0.01 | 0.5 | 0.5 | 15 | 10 | 10 | 5 |
| Pubmed | MLP-D | 0.01 | 0.01 | 0.25 | 0.25 | 15 | 100 | 5 | 5 |
| ogbn-arxiv | MLP | 0.01 | 0.001 | 0.25 | 0.5 | 15 | 10 | 1 | 5 |
| ogbn-arxiv | MLP-D | 0.01 | 0.001 | 0.5 | 0.25 | 15 | 10 | 1 | 5 |
| Wine | FP | 0.01 | 0.001 | 0.5 | 0.5 | 20 | 0.1 | 5 | 5 |
| Wine | MLP | 0.01 | 0.001 | 0.5 | 0.25 | 20 | 0.1 | 5 | 5 |
| Wine | MLP-D | 0.01 | 0.01 | 0.25 | 0.5 | 10 | 1 | 5 | 5 |
| Cancer | FP | 0.01 | 0.001 | 0.5 | 0.25 | 20 | 0.1 | 5 | 5 |
| Cancer | MLP | 0.01 | 0.001 | 0.5 | 0.5 | 20 | 1.0 | 5 | 5 |
| Cancer | MLP-D | 0.01 | 0.01 | 0.5 | 0.5 | 20 | 0.1 | 5 | 5 |
| Digits | FP | 0.01 | 0.001 | 0.25 | 0.5 | 20 | 0.1 | 5 | 5 |
| Digits | MLP | 0.01 | 0.001 | 0.25 | 0.5 | 20 | 10 | 5 | 5 |
| Digits | MLP-D | 0.01 | 0.001 | 0.5 | 0.25 | 15 | 0.1 | 5 | 5 |
| 20news | FP | 0.01 | 0.01 | 0.5 | 0.5 | 20 | 500 | 5 | 5 |
| 20news | MLP | 0.001 | 0.001 | 0.25 | 0.5 | 20 | 500 | 5 | 5 |
| 20news | MLP-D | 0.01 | 0.01 | 0.25 | 0.25 | 20 | 100 | 5 | 5 |
| MNIST (1000) | MLP | 0.01 | 0.01 | 0.5 | 0.5 | 15 | 10 | 10 | 5 |
| MNIST (2000) | MLP-D | 0.01 | 0.001 | 0.5 | 0.5 | 15 | 100 | 10 | 5 |
| MNIST (3000) | MLP | 0.01 | 0.01 | 0.5 | 0.5 | 15 | 10 | 5 | 5 |

used cosine). Then $\tilde{A} = \mathsf{kNN}(X') = M \odot S$ where $\odot$ represents the Hadamard (element-wise) product. With this formulation, in the forward phase of the network, one can first compute the matrix $M$ using an off-the-shelf k-nearest neighbors algorithm and then compute the similarities in $S$ only for pairs of nodes where $M_{ij} = 1$. In our experiments, we compute exact k-nearest neighbors; one can approximate it using locality-sensitive hashing approaches for larger graphs (see, e.g., [5, 7]). In the backward phase of our model, we compute the gradients only with respect to those elements in $S$ whose corresponding value in $M$ is 1 (i.e. those elements $S_{ij}$ such that $M_{ij} = 1$); the gradient with respect to the other elements is 0. Since $S$ is computed based on $X'$, the gradients flow to the elements in $X'$ (and consequently to the weights of the MLP) through $S$.

**Adjacency processor:** We used a function P in our adjacency processor to make the values of the $\tilde{A}$ positive. In our experiments, when using an MLP generator, we let P be the ReLU function applied element-wise on the elements of $\tilde{A}$. When using the fully-parameterized (FP) generator, applying ReLU results in a gradient flow problem as any edge whose corresponding value in $\tilde{A}$ becomes less than or equal to zero stops receiving gradient updates. For this reason, for FP we apply the ELU [3] function to the elements of $\tilde{A}$ and then add a value of 1.

## C  Dataset statistics

The statistics of the datasets used in the experiments can be found in Table 2.

# D   Supervision starvation in Erdős-Rényi and scale-free networks

We start by defining some new notation that helps simplify the proofs and analysis in this section. We let $l_v$ be a random variable indicating that $v$ is a labeled node, with $\overline{l_v}$ indicating that its negation, $c_{v,u}$ be a random variable indicating that $v$ is connected to $u$ with an edge, with $\overline{c_{v,u}}$ indicating its negation, and $cl_v$ be random variable indicating that $v$ is connected to at least one labeled node with $\overline{cl_v}$ indicating its negation (i.e. it indicates that $v$ is connected to no labeled nodes).

**Theorem 1** *Let $\mathcal{G}(n, m)$ be an Erdős-Rényi graph with $n$ nodes and $m$ edges. Assume we have labels for $q$ nodes selected uniformly at random. The probability of an edge being a starved edge with a two-layer GCN is equal to $(1 - \frac{q}{n})(1 - \frac{q}{n-1}) \prod_{i=1}^{2q} (1 - \frac{m-1}{\binom{n}{2}-i})$.*

**Proof 1** *To compute the probability of an edge being a starved edge, we first compute the probability of the two nodes of the edge being unlabeled themselves and then the probability of the two nodes not being connected to any labeled nodes. Let $v$ and $u$ represent two nodes connected by an edge.*

*With $n$ nodes and $q$ labels, the probability of a node being labeled is $\frac{q}{n}$. Therefore, $Pr(\overline{l_v}) = (1 - \frac{q}{n})$ and $Pr(\overline{l_u} \mid \overline{l_v}) = (1 - \frac{q}{n-1})$. Therefore, $Pr(\overline{l_v} \wedge \overline{l_u}) = (1 - \frac{q}{n})(1 - \frac{q}{n-1})$.*

*Since there is an edge between $v$ and $v$, there are $m - 1$ edges remaining. Also, there are $\binom{n}{2} - 1$ pairs of nodes that can potentially have an edge between them. Therefore, the probability of $v$ being disconnected from the first labeled node is $1 - \frac{m-1}{\binom{n}{2}-1}$. If $v$ is disconnected from the first labeled node, there are still $m - 1$ edges remaining and there are now $\binom{n}{2} - 2$ pairs of nodes that can potentially have an edge between them. So the probability of $v$ being disconnected from the second node given that it is disconnected from the first labeled node is $1 - \frac{m-1}{\binom{n}{2}-2}$. With similar reasoning, we can see that the probability of $v$ being disconnected from the $i$-th labeled node given that it is disconnected from the first $i - 1$ labeled nodes is $1 - \frac{m-1}{\binom{n}{2}-i}$.*

*We can follow similar reasoning for $u$. The probability of $u$ being disconnected from the first labeled node given that $v$ is disconnected from all $q$ labeled nodes is $1 - \frac{m-1}{\binom{n}{2}-q-1}$. That is because there are still $m - 1$ edges remaining and $\binom{n}{2} - q - 1$ pairs of nodes that can potentially be connected with an edge. We can also see that the probability of $u$ being disconnected from the $i$-th labeled node given that it is disconnected from the first $i - 1$ labeled nodes and that $v$ is disconnected from all $q$ labeled nodes is $1 - \frac{m-1}{\binom{n}{2}-q-i}$.*

*As the probability of the two nodes being unlabeled and not being connected to any labeled nodes in the graph are independent, their joint probability is the multiplication of their probabilities computed above and it is equal to $(1 - \frac{q}{n})(1 - \frac{q}{n-1}) \prod_{i=1}^{2q} (1 - \frac{m-1}{\binom{n}{2}-i})$.*

**Barabási–Albert and scale-free networks:** We also extend the above result for Erdős-Rényi graphs to the Barabási–Albert [1] model. Since Barabási–Albert graph generation results in scale-free networks with a scale parameter $\gamma = -3$, we present results for the general case of scale-free networks as it makes the analysis simpler and more general. In what follows, we compute the probability of an edge being a starved edge in a scale-free network.

Let $\mathcal{G}$ be a scale-free network with $n$ nodes, $q$ labels (selected uniformly at random), and scale parameter $\gamma$. Then, if we select a random edge between two nodes $v$ and $u$, the probability of the edge between them being a starved edge is:

$$Pr(\overline{l_v}) * Pr(\overline{l_u}|\overline{l_v}) * Pr(\overline{cl_v}|c_{v,u}, \overline{l_v}, \overline{l_u}) * Pr(\overline{cl_u}|c_{v,u}, \overline{l_v}, \overline{l_u}, \overline{cl_v}).$$

Each of these terms can be computed as follows ($\binom{a}{b}$ represents the number of combinations of selecting $b$ items from a set with $a$ items):

- $Pr(\overline{l_v}) = (1 - \frac{q}{n})$
- $Pr(\overline{l_u}|\overline{l_v}) = (1 - \frac{q}{n-1})$

Table 2: Dataset statistics.

| Dataset | Nodes | Edges | Classes | Features | Label rate |
|---------|-------|-------|---------|----------|------------|
| Cora | 2,708 | 5,429 | 7 | 1,433 | 0.052 |
| Citeseer | 3,327 | 4,732 | 6 | 3,703 | 0.036 |
| Pubmed | 19,717 | 44,338 | 3 | 500 | 0.003 |
| ogbn-arxiv | 169,343 | 1,166,243 | 40 | 128 | 0.537 |
| Wine | 178 | 0 | 3 | 13 | 0.112 |
| Cancer | 569 | 0 | 2 | 30 | 0.035 |
| Digits | 1,797 | 0 | 10 | 64 | 0.056 |
| 20news | 9,607 | 0 | 10 | 236 | 0.021 |
| MNIST | 10,000 | 0 | 10 | 784 | 0.1, 0.2 and 0.3 |

- $Pr(\overline{cl_v}|c_{v,u}, \overline{l_u}, \overline{l_v}) = \dfrac{\sum_{k=1}^{n-1} k^\gamma \frac{\binom{n-q-2}{k-1}}{\binom{n-2}{k-1}}}{\sum_{k=1}^{n-1} k^\gamma}$

For a large enough network, $Pr(\overline{cl_u}|c_{v,u}, \overline{l_v}, \overline{l_u}, \overline{cl_v})$ can be approximated as $Pr(\overline{cl_u}|c_{v,u}, \overline{l_v}, \overline{l_u})$ and it can be computed similarly as the previous case.

With the derivation above, for a scale-free network with $n = 2708$ and $q = 140$ (corresponding to the stats from Cora), the probability of an edge being a starved edge for $\gamma = -3$ is $0.87$ and for $\gamma = -2$ is $0.76$ .

# E    Why not compare the learned graph structures with the original ones?

A comparison between the learned graph structures using SLAPS (or other baselines) and the original graph structure of the datasets we used may not be sensible. We explain this using an example. Before getting into the example, we remind the reader that the goal of structure learning for semi-supervised classification with graph neural networks is to learn a structure with a high degree of homophily. Following [17], we define the *edge homophily ratio* as the fraction of edges in the graph that connect nodes that have the same class label.

Figure 4 demonstrates an example where two graph structures for the same set of nodes have the same edge homophily ratio (0.8 for both) but have no edges in common. For our task, it is possible that the original graph structure (e.g., the citation graph in Cora) corresponds to the structure on the left but SLAPS (or any other model) learns the graph on the right, or vice versa. While both these structures may be equally good[3], they do not share any edges. Therefore, measuring the quality of the learned graph using SLAPS by comparing it to the original graph of the datasets may not be sensible. However, if a noisy version of the initial structure is provided as input for SLAPS, then one may expect that SLAPS recovers a structure similar to the cleaned original graph and this is indeed what we demonstrate in the main text.

# F    Limitations

In this section, we discuss some of the limitations of the proposed model. Firstly, in cases where nodes do not have input features but an initial noisy structure of the nodes is available, our self-supervised task cannot be readily applied. One possible solution is to first run an unsupervised node embedding model such as DeepWalk [13] to obtain node embeddings, then treat these embeddings as node features and run SLAPS. Secondly, the FP graph generator is not applicable in the inductive setting; this is because FP directly optimizes the adjacency matrix. However, our other two graph generators (MLP and MLP-D) can be applied in the inductive setting.

---

[3]We are disregarding the features for simplicity sake.

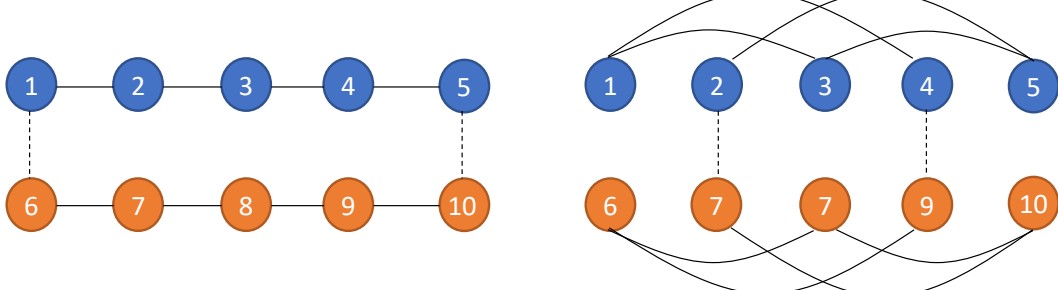

Figure 4: Two example graph structures. Node colors indicates the class labels. Solid lines indicate homophilous edges and dashed lines indicate non-homophilous edges. The two graphs exhibit the same degree of homophily yet there is not overlap between their edges.