# OpenReview forum: "SLAPS: Self-Supervision Improves Structure Learning for Graph Neural Networks"
_NeurIPS.cc/2021/Conference — NeurIPS 2021 Poster_

### Official Review · Reviewer_QmgH · 2021-07-14

**Rating:** 6
**Confidence:** 4

**Summary:**

This paper studies graph structure learning. Given a dataset with latent but unobserved graph structure, the goal is to infer the graph structure and help downstream tasks. The authors specifically design a feature reconstruction task as self-supervision to improve the structure learning.

**Limitations And Societal Impact:**

Would be better to add more experiments to support the effectiveness of feature reconstruction task as I described above.

**Main Review:**

Overall I think the direction of using self-supervised learning for learning latent graph structure is very promising, especially for the cases that the authors point out, when the training signal is sparse and cannot propagate to many edges via a local GNN. The feature reconstruction is obviously a good starting point and reasonable choice.

For the experiments, I have several questions and concerns:

(1) For experiment 5.3, the authors say they add noisy graph as input. But I think the assumption of structure learning is we don't observe any graph. Is this experiment aimed at testing how well the SLAPS can correct the perturbed edges? If so, I think it would be more fair to compare with the other graph structure learning baselines.

(2) Experiment 5.4.1 is very interesting and shows that SLAPS could recover the ground-truth graph structure. However the experiment is conducted in that edges is randomly removed or added. A more realistic setting is that we have a set of nodes without any link to it, which is more close to your analysis about (starved edges). It would be more convincing if the authors could conduct such experiments (for example, randomly select K nodes and delete all the internal edges, or first run a cluster algorithm and pick one cluster to delete, or if you have timestamps, just pick the node after a particular timestamp). Also, it would be intersting to see some visualization results.

(3) For experiment, I think it would be better to also include "Graph Structure Learning for Robust Graph Neural Networks" as a baseline (their regularization could also be regarded as a self-supervised task). Also, it would be better to add a GCN results with groundtruth graph structure for comparison (as upper bound)

(4) The authors mainly show the results for node classification, where similar nodes will have tendency to be connected. What about other dataset, such as molecules? Is the feature reconstruction also beneficial to reconstruct a molecule structure? It would be interesting to also test it.

(5) The notation is a little bit confusing to me in Table 1 (I previously thought self-training is L_DAE). It would be better to also include an ablation that remove the feature reconstruction (L_DAE) in the main table to support your hypothesis.

Overall, I think the paper is well written and a good starting point in this research direction.

**Time Spent Reviewing:**

2

---

> ### Author Response · Authors · 2021-08-08
> **Response**
>
> We thank the reviewer for their insightful feedback and encouraging words regarding the direction of our paper. We start with responding to concern (5) of the reviewer which has also been mentioned in the limitations section of the review. Then we will proceed to the other concerns.
>
> **(5) ablation that removes the feature reconstruction (L_DAE)** The paper contains multiple ablation studies for this. Removing the feature reconstruction (L_DAE) is achieved by setting $\lambda=0$. The first ablation study is in lines 326-336 (Figure 4) where we show the performance for various values of $\lambda$, including $\lambda=0$. The second ablation study is in the experiment in lines 349-364 (Figure 5) where we compare SLAPS($\lambda>0$) with SLAPS($\lambda=0$), i.e. SLAPS with and without the feature reconstruction task.
>
> Regarding the confusion, we would like to emphasize the difference between self-supervision and self-training: self-supervision is the task we proposed in section 4.5 and corresponds to $\mathcal{L}_{DAE}$. Self-training is a technique previously proposed in the literature and explained in the supplementary (line 108). We apologize for the high similarity of the two names which causes confusion; both names come from the existing literature and we decided to not change them.
>
> **(1,2) Experiment 5.3 and Experiment 5.4.1:**
> Experiments 5.3 and 5.4.1 are mainly meant to provide a proxy for measuring the quality of the graph structure learned by SLAPS (experiment 5.3 provides the grounds for the later experiment in 5.4.1). They are meant to be considered as small-scale ablation studies as opposed to a large-scale evaluation of SLAPS in a different problem setting which, as the reviewer pointed out, requires a much bigger empirical evaluation. Thanks for pointing this out, we will clarify it in our next revision.
>
> **Regarding the random edge removal strategy:** We would like to clarify that starved edges do not occur only for nodes that are connected to **no other nodes**. It occurs for nodes that are connected to **no other labeled nodes** which is a much looser condition. In fact, with 25% random perturbation of the original graph structure in Cora in sections 5.3 and 5.4.1, on average (of 5 runs) 50.6% of the edges are starved edges. For this reason, we believe the setting in this experiment is indeed realistic and close to our analysis of the starved edges. However, if the reviewer still believes that the suggested experiments could provide more insight, please let us know and we will conduct the suggested experiment.
>
> **(3) The regularization of "Graph Structure Learning for Robust Graph Neural Networks":** The regularization in the suggested paper is in the same vein as (although not identical to) the one we test in line 53 of the supplementary, as well as the regularization used in the IDGL paper. As the results indicate, both models are outperformed by SLAPS that leverages the feature reconstruction task.
>
> **(3) GCN results with ground-truth graph structure as upper bound:** We will do this in our next revision. Thanks for the suggestion. We would like to mention that for some datasets, GCN with the ground truth graph structure is even outperformed by SLAPS (e.g., GCN with the ground truth graph structure achieves 70.3% accuracy on Citeseer whereas SLAPS achieves 72.6% accuracy).
>
> **(4) Extension to molecule datasets:** In this paper, we mainly focused on node classification which is one of the biggest areas where GNNs have shined. For molecule classification (which is an instance of the graph classification problem), if we assume a graph structure is not available (which is the main problem setup of our paper), then all that is left is a set of atom types that may not contain enough information for classification regardless of the method.

---

> > ### Comment · Reviewer_QmgH · 2021-08-23
> > **Comments after rebuttal**
> >
> > Thanks authors for the efforts to answer my questions. Most of them has been addressed.
> >
> > Regarding to the edge removal experiments, I understand authors' claim that starved edges are those cases that nodes are not connected to labelled nodes. I just think such random perturbation is not that realistic in real-world problem. I still think it would be more convincing to add a more strict perturbation, and also provide some visualization (such as what kind of edges are recovered)
> >
> > Overall, I think this paper provides a good starting point on this research direction, so I still lean towards acceptance.

---

> ### Author Response · Authors · 2021-08-18
> **Has our response addressed your concerns?**
>
> Dear reviewer,
>
> We would be grateful if you can confirm whether our response has addressed your concerns, and let us know if any issues remain.

---

### Official Review · Reviewer_KpLH · 2021-07-15

**Rating:** 6
**Confidence:** 4

**Summary:**

The paper first identifies a starved edge problem in latent graph learning. That is, nodes that are far from labeled nodes may receive insufficient supervision signal during learning. To address this issue, the paper then proposes to leverage a self-supervision signal, which comes from denoising the node features of the graph. Through taking both the noisy node features and the currently learned adjacency matrix as input, the feature denoising process acts as a regularizer to the learning of the adjacency matrix. Experiments on benchmark tasks show that the proposed strategy outperforms several comparison baseline models with or without adjacency matrices given.

The identification of the starved edge problem in latent graph learning is interesting. The proposed solution that introduces inductive bias in adjacency matrix learning through leveraging node features makes sense to me. Nevertheless, I found that the impact of the starved edge problem to a graph model’s predictive performance could be better justified with stronger evidence and the empirical studies in its current form could be further improved.


**Limitations And Societal Impact:**

yes

**Main Review:**

The paper is well written and easy to follow. The related work section is informative and concise; the ablation studies provided in Sections 5.2-5.4 are beneficial.

Graph learning in data where graph structure is not readily available is an interesting and important research problem. Also, leveraging node features to regularize the learning of the adjacency matrix seems novel and makes sense to me. Nevertheless, I have the following two main concerns regarding the paper.

1.  Stronger comparison baselines may be beneficial. The proposed model has a larger number of parameters than that of other comparison baseline models. In this sense, I am not sure where is the predictive improvement really coming from. That is, how much the accuracy improvement benefits from extra model freedoms/parameters? For example, the current baseline MLP is a very simple approach. To make the comparison more convincing, a stronger baseline may be useful, such as a Transformer Encoder (with attention) as a non-graph structure baseline for comparison.

2.  I did not see a direct connection between the predictive improvement and addressing the supervision starvation problem (i.e., starved edges). It is useful to identify and show the existence of the starved edges problem in common graph datasets. But the importance of this issue, in terms of its impact on the predictive performance of the trained model, is not clear to me. I think it would be useful to show the importance of the starved edges directly since this is one of the main contribution claims of the paper. This is particularly relevant since as stated in Lines 297-298 that adding AdaEdge does not help. The paper suggests that such non-improvement may due to the proposed method has already learned a good graph structure. On the other hand, I think this might also indicate that some graph structures are just not important to the predictive targets at all.

Minor comments:

1.  Lines 188-189: for real-world graphs, how the percentage of starved edges was calculated is not clear to me.
2.  Line 390: the statement of “learning the graph structure of the nodes simultaneously based on self-supervision” is a bit ambiguous to me. The self-supervision here is more like an extra task and just act as a model regularizer. It would be useful to make the statement more precise.

==== post-rebuttal update ====

I thank the authors for their response to my reviews; really appreciate that!
I do not have any major concerns regarding the paper, and hope some of the experiments/analyses from the rebuttal can be added to the main text of the paper in revision.



**Time Spent Reviewing:**

6

---

> ### Author Response · Authors · 2021-08-08
> **Response**
>
> We thank the reviewer for insightful feedback, especially for the suggested set of experiments and comparisons.
>
> **Stronger comparison baseline: benefits from extra model freedoms/parameters:**
> We provide four pieces of evidence showing the improvement is mainly coming from addressing the problem with the starved edges as opposed to extra parameters.
> * For the first piece of evidence, please see our response to your question on directly measuring the importance of the starved edges.
> * Second, the experiment starting at line 47 in the supplementary material fixes a prior graph manually instead of using the proposed self-supervision. Since the prior graph adds no extra parameters, the number of parameters for this model is within the same range as our baselines. However, this model still outperforms the baselines.
> * Third, since the extra parameters in SLAPS are used for the self-supervised task, our GCN model has the same capacity as that of the baselines. That is, we are comparing GCN classifiers with the same learning capacity and so the improvement should be coming from learning a better graph structure.
> * Fourth, some of our model variants have fewer parameters than the baselines. As an example, for the Cora dataset, besides the parameters for the GCN_C model, our SLAPS + MLP and SLAPS + MLP-D models have $2 * 1433 * 1433 + 2 * 1433 * 512 = 5.7M$ and $2 * 1433 + 2 * 1433 * 512=1.5M$ extra parameters respectively, whereas LDS has $2708 * 2708 = 7.3M$ extra parameters.
>
> **Stronger comparison baseline: Transformer encoder:**
> Upon your request and reviewer 5ygi, we conducted the following experiments.
> * We tested a fully connected GCN and observed that the model always learns to predict the majority class. This is expected because after one fully connected GCN layer, all nodes will have the same embedding and become indistinguishable.
> * We also experimented with fully connected GAT and observed similar behaviour. We believe this is because the attention weights are (almost) random at the beginning (due to random initialization of the model parameters) resulting in nodes becoming indistinguishable and GAT cannot escape from that state.
> * Besides GCN and GAT, as suggested by the reviewer, we also experimented with a Transformer encoder. The skip connections of Transformer helped avoid the problem observed for GCN and GAT and we were able to achieve better results (~40% accuracy on Cora). However, we observed severe overfitting (even with very small models and with high dropout probabilities). This may be because Transformer learns proper attention weights for the nodes in the training, but not for those in the validation and test. This is another testament to the “starved edge” problem we identified.
> * We also ran one more experiment where instead of using a fully connected graph, we started with a kNN graph with a large value for k in the hope that the initial kNN graph has a high recall and the attention weights can prune the irrelevant edges and increase precision. We were able to obtain an accuracy of about 62% with this approach which is still far behind the results of our model.
>
> **Showing the importance of the starved edges directly:**
> Besides the evidence provided in the paper, to measure the importance of the starved edges more directly, we conducted the following experiment during the rebuttal. We compared SLAPS with and without self-supervision on two groups of test nodes on Cora: 1) those that are not connected to any labeled nodes after training, and 2) those that are connected to at least one labeled node after training. The nodes in group one have a high chance of having starved edges. We observed that adding self-supervision provides $38.0%$ improvement for the first group and only $8.9%$ improvement for the latter. Since self-supervision mainly helps with nodes in group 1, this provides evidence that the starved edges are problematic and the improvement mainly comes from addressing that issue.
>
> We would like to thank the reviewer for bringing this to our attention; we believe the results discussed above provide more intuition and make our claim stronger.
>
> **Minor comments: starved edges calculation for real-world graphs:** We counted the percentages of edges whose two nodes are not labeled and not connected to any other labeled nodes.
>
> **Minor comments: the statement in line 390:** Thanks. We will revise the text. The goal is to show that the graph structure of the nodes is learned both by the downstream task and the self-supervised task.

---

> > ### Comment · Reviewer_KpLH · 2021-08-25
> > **No major concerns**
> >
> > Thank you for your response to my reviews; really appreciate that.
> >
> > I do not have any major concerns at this point, and I hope some of the above experiments/analyses could be added to the main text of the paper.

---

> ### Author Response · Authors · 2021-08-18
> **Has our response addressed your concerns?**
>
> Dear reviewer,
>
> We would be grateful if you can confirm whether our response has addressed your concerns, and let us know if any issues remain.

---

### Official Review · Reviewer_5ygi · 2021-07-16

**Rating:** 8
**Confidence:** 5

**Summary:**

The authors propose SLAPS, a methodology for deriving latent k-NN graphs for usage as edges of a GNN. They argue theoretically and empirically that it is important to combine self-supervised learning with graph structure learning, to avoid the prevalent edge starvation problem. They validate an approach consisting of a denoising self-supervised objective and various graph derivation strategies against relevant latent graph inference benchmarks, achieving impressive results.

**Limitations And Societal Impact:**

No concerns.

**Main Review:**

I found the paper well-written, easy to follow and full of useful intuition.

The authors re-use a familiar construct for latent graph inference (the k-NN graph), and argue well for addition of a self-supervised loss when inferring it. This combination is novel in principle, and the obtained results appear significant.

The impact for GNN practitioners is clear. In many cases the GNN datasets will contain suboptimal graphs---or complete lack thereof---and different stages of processing may require a different graph. Besides the obvious applicability for relational reasoning, there is potential for spillover into set-structured tasks, or even sequence-style tasks, making the work relevant to a wider community.

Further, I really appreciate that the authors hadn't merely 'stitched' a self-supervised loss to the architecture without any justification. This choice was substantiated with clear theoretical arguments (edge starvation) as well as a workable hypothesis for denoising to be used. The qualitative analysis (e.g. on homophily properties) is also a nice touch. I recommend acceptance.

I have some suggestions for follow-up experiments that may strengthen the paper further:
- The theoretical expected ratio of starved edges is currently provided only for Erdős-Rényi graphs, which are purely uniformly distributed and may not be very naturalistic. I recommend the authors also expand their analysis to Barabási-Albert graphs, which might be more appropriate for many complex naturalistic phenomena.
- Could the authors also include a fully-connected graph baseline (i.e. where all nodes are mutually connected), using a model such as GAT or MPNN? I find such baselines to always be insightful about the impact of graph inference.
- The investigated latent graph inference baselines could be expanded to also include the Differentiable Graph Module (Kazi et al.), which attacks the "added supervision" problem via reinforcement learning-style objectives.

==== Post-rebuttal update: I thank the authors for responding to my comments carefully and faithfully. The theoretical result on Barabási-Albert graphs will surely improve the paper. I have upgraded my score as a result. Well done!

**Time Spent Reviewing:**

3

---

> ### Author Response · Authors · 2021-08-08
> **Response**
>
> We thank the reviewer for valuable feedback and for pointing out the exciting potential for the applicability of our work to set-structured and sequence-style tasks. We have been indeed thinking about the latter task and have ideas for extending this work to those tasks.
>
> **Extending Theorem 1 to Barabasi-Albert Graphs:**
> We post the proof and the results for extending Theorem 1 to Barabasi-Albert graphs in a separate response to keep this response short. Thanks for bringing this to our attention; we believe the extended theorem makes our claim stronger.
>
> **Experiments with fully connected graphs:**
> * We tested a fully connected GCN and observed that the model always learns to predict the majority class. This is expected because after one fully connected GCN layer, all nodes will have the same embedding and become indistinguishable.
> * We also experimented with fully connected GAT and observed similar behaviour. We believe this is because the attention weights are (almost) random at the beginning (due to random initialization of the model parameters) resulting in nodes becoming indistinguishable and GAT cannot escape from that state.
> * Besides GCN and GAT, we also experimented with a Transformer encoder. The skip connections of Transformer helped avoid the problem observed for GCN and GAT and we were able to achieve better results (~40% accuracy on Cora). However, we observed severe overfitting (even with very small models and with high dropout probabilities). This may be because Transformer learns proper attention weights for the nodes in the training, but not for those in the validation and test. This is another testament to the “starved edge” problem we identified.
> * We also ran one more experiment where instead of using a fully connected graph, we started with a kNN graph with a large value for k in the hope that the initial kNN graph has a high recall and the attention weights can prune the irrelevant edges and increase precision. We were able to obtain an accuracy of about 62% with this approach which is still far behind the results of our model.
>
> **Deep Graph Module:**
> Unfortunately, we were not able to find their code. We will, however, discuss their work in our next revision.

---

> > ### Author Response · Authors · 2021-08-08
> > **Barabasi-Albert Graphs**
> >
> > Since Barabasi-Albert graph generation results in scale-free networks with a scale parameter $\gamma=-3$, we present a proof for the general case of scale-free networks as it makes the proof simpler and more general.
> >
> > Let G be a scale-free network with N nodes, Q labels (selected uniformly at random), and scale parameter $\gamma$. Then, if we select a random edge between two nodes V and U, the probability of the edge between them being a starved edge is:
> > * P(V is not labeled) * P(U is not labeled | V is not labeled) * P(V is connected to no labeled nodes | V is connected to U, V is not labeled, U is not labeled) * P(U is connected to no labeled nodes | V is connected to U, V is not labeled, U is not labeled, V is connected to no labeled nodes).
> >
> > Each of these terms can be computed as follows ($C(a, b)$ represents the number of combinations of selecting b items from a set with a items):
> > * P(V is not labeled) = $(1 - Q/N)$
> > * P(U is not labeled | V is not labeled) = $(1-Q/(N-1))$
> > * P(V is connected to no labeled nodes | V is connected to U, V is not labeled, U is not labeled) = $\frac{\sum_{k=1}^{n-1} k^{\gamma} \frac{C(N-Q-2, k-1)}{C(N-2, k-1)}}{\sum_{k=1}^{n-1} k^{\gamma}}$.
> > * For a large enough network, P(U is connected to no labeled nodes | V is connected to U, V is not labeled, U is not labeled, V is connected to no labeled nodes) can be approximated as P(U is connected to no labeled nodes | V is connected to U, V is not labeled, U is not labeled) and the computation is similar to the previous case.
> >
> > With the derivation above, for a scale-free network with $N=2708$ and $Q=140$ (corresponding to the stats from Cora), the probability of an edge being a starved edge for $\gamma=-3$ is $0.87$ and for  $\gamma=-2$ is $0.76$.

---

> > ### Comment · Reviewer_5ygi · 2021-08-08
> > **Score upgrade**
> >
> > Thank you for your careful and detailed comment, and the new theoretical results.
> >
> > I have upgraded my score to an 8. Good luck!

---

### Official Review · Reviewer_5rQg · 2021-07-19

**Rating:** 5
**Confidence:** 4

**Summary:**

This paper combines the self-supervised learning with graph structure learning. A model named SLAPS is proposed to alleviate the "supervision starvation" problem of structure learning methods by denoising features using generated graph. Extensive experiments are conducted to show the effectiveness of the model.

**Limitations And Societal Impact:**

Limitations are discussed in the supplementary materials.

**Main Review:**

The major strong points are:

S1. The idea of combining SSL to GSL is interesting. To my knowledge, it might be the first work that combines GSL with self-supervised learning.

S2. A model SLAPS is proposed, and appears to be effective under certain settings.

Nevertheless, there are also several weak points:

W1. Arguable claims: The major argument of supervision starvation is arguably wrong. In fact, the so-called "starved edges" can be optimized by most graph structure learning methods based on metric learning. Specifically, the classifier, i.e. GNN_C in the paper, is a function of \tilde{A}, as the graph learning module models the \tilde{A} using graph learning parameters, it can be optimized.
Moreover, many graph structure learning models apply graph regularization terms. For example, a feature smoothness loss [Chen et al., 2020] can be somehow viewed as a self-supervised term that regularizes the model by homophily assumption, which also provides self-supervision for "starved_edges". Besides, other graph regularization terms introduced in [Wei et al., 2020] can also somehow provide supervision for all the potential edges.

W2. Model design: The defined graph structure learning is solely based on the input feature, which overlooks the adjacency matrix. Actually, the optimal adjacency matrix is mostly a shift [Zhao et al., 2021] from the original adjacency matrix. Neglecting the graph structure and generate it via features only apparently hampers the performance.

W3. Experiments: The experimental setting is problematic. For models requiring an initial graph structure, the adjacency model is completely abandoned and instead replaced by a kNN graph. This setting is quite unfair to models that take both adjacency matrix and node features. For example, on Cora, IDGL should be around 84.5; however, with kNN graph as input, the reported results are around 70.9.

Besides, the IDGL model actually is able to scale to OGB-Arxiv with a scalable version IDGL-Anch in the NIPS version [Chen et al., 2020] (The authors seem to be unaware of the NIPS version and cites the AAAI workshop version).

W4. Other minor issues:
- Grale [Halcrow et al., 2020], which also takes the features as input only, should also be compared. It is also interesting to compare Grale with SLAPS since Grale does not consider self-supervision. However, it is understandable since the code for Grale is unavailable.

- References require revision: Some conference papers are cited with their arxiv version. NIPS version of IDGL should be cited.
[Wei et al., 2020] Graph Structure Learning for Robust Graph Neural Networks. In KDD, 2020.
[Chen et al., 2020] Iterative Deep Graph Learning for Graph Neural Networks- Better and Robust Node Embeddings. In NIPS, 2020.
[Zhao et al., 2021] Data Augmentation for Graph Neural Networks. In AAAI, 2021.
[Halcrow et al., 2020] Grale: Designing networks for graph learning. In KDD, 2020.


**Time Spent Reviewing:**

10

---

> ### Author Response · Authors · 2021-08-08
> **Response**
>
> We thank the reviewer for constructive feedback and encouraging comments regarding the combination of self-supervised learning and graph structure learning.
>
> **Clarifying a possible misunderstanding of the problem setup:**
> We believe there might be a misunderstanding about the problem we are studying in this paper. So before answering the specific comments, we would like to clarify the problem setup.
> As mentioned in line 2 of the abstract and lines 19-20 of the introduction, our goal is to develop GNN models for domains where a graph structure is **NOT** available. This is different from the problem setup where the graph structure is available and the goal is to refine it. The problem we study is ubiquitous in industrial applications and extends the applicability of GNNs to a large domain of problems.
> With this clarification, we first start with answering W2 and W3 and then move on to W1 and W4.
>
> **W2 & W3: Model design and Experiments:**
> We hope the clarification provided above resolves these two comments. We feed a kNN to the baselines because we assume an initial graph structure is not available in our problem setup. We are not advocating for using a kNN structure instead of the initial structure in problems where an initial structure is available.
>
> Note: The experimental setup is completely fair because we also initialize our SLAPS generators with the kNN graph.
>
> **W1: arguable claims:**
> It is somewhat unclear why the reviewer thinks this is an arguable claim. We appreciate it if the reviewer points out which part of Theorem 1 (which formally establishes our claim) is arguable. In what follows, we provide answers to the individual comments and provide more evidence on why starved edges are problematic.
> * **Approaches based on metric learning:** Take Figure 2 in our paper as an example. Let V and U represent the two nodes on the left. Let M1 and M2 be two models that project the nodes to a metric space in such a way that they: 1) both induce the solid lines in Figure 2, 2) M1 also induces an edge between V and U, but 3) M2 does not. Then, since the edge between U and V does not affect the loss function, during training both M1 and M2 receive the same loss. However, M1 and M2 may have very different behaviours at the test time when predictions are to be made for other nodes in the graph. This is what we call “starvation”: the edge between U and V receives no supervision in this case. We would also like to point out that SLAPS is also a metric learning-based approach: the MLP generator projects the nodes into a metric space where closeness corresponds to the existence of edges.
> * **GNN_C is a function of \tilde{A} and \tilde{A} can be optimized:** \tilde{A} is only partially optimized. More specifically, only those parts of \tilde{A} that play a direct role in predicting the training nodes can be optimized. The other parts of \tilde{A} do not receive enough supervision to be properly optimized and this results in low performance on the validation/test sets.
> * **Other regularizations:** We indeed explained this in lines 207-210 of the main text and provided experiments and deep-dive analyses in lines 47-81 of the supplementary.
> * **One more piece of evidence:** As one more piece of evidence regarding how the starved edges are problematic, we conducted the following experiment during the rebuttal. We compared SLAPS with and without self-supervision on two groups of test nodes on Cora: 1) those that are not connected to any labeled nodes after training, and 2) those that are connected to at least one labeled node after training. The nodes in group one have a high chance of having starved edges. We observed that adding self-supervision provides 38.0% improvement for the first group and only 8.9% improvement for the latter. Since self-supervision mainly helps with nodes in group 1, this provides evidence that the starved edges are problematic and the improvement mainly comes from addressing that issue.
>
> **W4. Other minor comments:**
> * **Grale:** As the reviewer mentioned, the code for Grale is unfortunately not available. We would like to point out that Grale is conceptually an extension of our kNN-GCN baseline and it can also benefit from the self-supervised task we proposed.
> Please also refer to Figs 4 and 5 in the paper for a comparison of models with and without the self-supervised task (\lambda=0 corresponds to no self-supervision).
> * **References require revision:** Thanks! We will update the references to reflect their latest version. We will also point out that IDGL-Anch scales to ogbn-arxiv.

---

> ### Author Response · Authors · 2021-08-18
> **Has our response addressed your concerns?**
>
> Dear reviewer,
>
> We would be grateful if you can confirm whether our response has addressed your concerns, and let us know if any issues remain.

---

### Decision · Program_Chairs · 2021-09-27

**Decision:**

Accept (Poster)

**Comment:**

This paper proposed an approach for simultaneously inferring graph structure and GNN parameters. In particular, the authors pointed out the problem of edge starvation when learning on graphs with limited number of labels and proposed a self-supervised approach to alleviate such an approach. Experimental results prove the effectiveness of the proposed approach on existing latent graph inference benchmarks.

Overall, the reviewers like the simplicity of the proposed approach with good intuitions by pointing out the edge starvation problem. The authors addressed most of the concerns raised by the reviewers.